# Tick-Borne Encephalitis Virus: A Quest for Better Vaccines against a Virus on the Rise

**DOI:** 10.3390/vaccines8030451

**Published:** 2020-08-12

**Authors:** Mareike Kubinski, Jana Beicht, Thomas Gerlach, Asisa Volz, Gerd Sutter, Guus F. Rimmelzwaan

**Affiliations:** 1Research Center for Emerging Infections and Zoonoses (RIZ), University of Veterinary Medicine Hannover, Foundation (TiHo), Buenteweg 17, 30559 Hannover, Germany; mareike.kubinski@tiho-hannover.de (M.K.); jana.beicht@tiho-hannover.de (J.B.); thomas.gerlach@tiho-hannover.de (T.G.); 2Institute of Virology, University of Veterinary Medicine Hannover, Foundation (TiHo), Buenteweg 17, 30559 Hannover, Germany; asisa.volz@tiho-hannover.de; 3Institute for Infectious Diseases and Zoonoses, Ludwig-Maximilians-University (LMU) Munich, Veterinaerstr. 13, 80539 Munich, Germany; gerd.sutter@lmu.de

**Keywords:** tick-borne encephalitis virus, TBEV, flavivirus, vaccination, vaccine, immunity, antibodies, CD4^+^ T cells, CD8^+^ T cells

## Abstract

Tick-borne encephalitis virus (TBEV), a member of the family *Flaviviridae*, is one of the most important tick-transmitted viruses in Europe and Asia. Being a neurotropic virus, TBEV causes infection of the central nervous system, leading to various (permanent) neurological disorders summarized as tick-borne encephalitis (TBE). The incidence of TBE cases has increased due to the expansion of TBEV and its vectors. Since antiviral treatment is lacking, vaccination against TBEV is the most important protective measure. However, vaccination coverage is relatively low and immunogenicity of the currently available vaccines is limited, which may account for the vaccine failures that are observed. Understanding the TBEV-specific correlates of protection is of pivotal importance for developing novel and improved TBEV vaccines. For affording robust protection against infection and development of TBE, vaccines should induce both humoral and cellular immunity. In this review, the adaptive immunity induced upon TBEV infection and vaccination as well as novel approaches to produce improved TBEV vaccines are discussed.

## 1. Tick-Borne Encephalitis Virus

Tick-borne encephalitis virus (TBEV), a member of the family *Flaviviridae* and genus Flavivirus [1], is one of the most important tick-transmitted pathogens in Europe and Asia, causing annually over 10,000 clinical cases [2]. The genus Flavivirus comprises several human-pathogenic arthropod-borne viruses such as yellow fever virus (YFV), dengue virus (DENV), Japanese encephalitis virus (JEV), Zika virus (ZIKV) and West Nile virus (WNV). Since TBEV is neurotropic, it can infect the central nervous system (CNS) leading to several neurological outcomes summarized as tick-borne encephalitis (TBE) (reviewed in [3]).

Mature TBE virions are approximately 50 nm in diameter and have an envelope consisting of membrane (M) and envelope (E) proteins anchored in a lipid bilayer. The nucleocapsid is composed of capsid (C) proteins and the RNA genome. The non-segmented, single-stranded RNA in positive orientation has one open reading frame (ORF) coding for a single polyprotein. This protein is co- and post-translationally cleaved by viral and host proteases into three structural proteins (C, precursor-M (prM), E) and seven non-structural proteins (NS1, NS2A, NS2B, NS3, NS4A, NS4B, NS5) [1,4,5]. As a viral surface glycoprotein, the E protein mediates receptor binding and membrane fusion of the viral and endosomal membrane. Moreover, it is important for inducing protective immunity [4,5,6]. The prM protein forms heterodimers with the E protein, thereby protecting the E protein fusion loop from premature fusion during flavivirus release [7]. In the *trans*-Golgi network, prM is cleaved by furin into pr and M [8,9] triggering the rearrangement of E proteins on the viral surface which leads to the transition from immature to mature virions (reviewed in [10]). The non-structural proteins of TBEV play an important role in replication, processing of the polyprotein and modulation of host cell functions (reviewed in [4]). Intracellular NS1 proteins are involved in the viral RNA replication (reviewed in [6]). However, NS1 is also secreted into the extracellular space as an oligomeric “soluble antigen” [11] and contributes to a protective immune response. NS3 is the viral serine protease (with NS2B as a co-factor), RNA helicase and nucleoside triphosphatase, therefore, having a central function in viral replication and protein processing. The highly conserved NS5 protein functions as the RNA-dependent RNA polymerase and methyltransferase. NS2A, NS4A and NS4B are presumably part of the replication complex. In addition, most of the non-structural proteins of TBEV are involved in immunomodulatory processes (reviewed in [6]).

TBEV is mainly transmitted to humans and animals via tick bites (reviewed in [12]). Occasionally, alimentary transmission after consumption of raw milk or dairy products of viremic sheep, cows or goats is also possible [13,14,15,16,17]. Occurrence of TBEV correlates with the distribution of its vector ticks, mainly *Ixodes ricinus* and *Ixodes persulcatus*, and ranges from Europe to Siberia, Russia and Far-Eastern countries (reviewed in [18]). Phylogenetic studies based on the E protein revealed three TBEV subtypes: European (TBEV-Eu), Siberian (TBEV-Sib) and Far-Eastern (TBEV-FE). However, two potential new subtypes were described: Himalayan (TBEV-Him) and Baikalian (TBEV-Bkl) [19,20]. During the last decades, the incidence of TBE has been fluctuating annually with a general upward trend in several European countries (reviewed in [21]). Additionally, TBEV and its vectors have invaded novel regions and countries, such as the Netherlands and the United Kingdom, as well as higher altitudes observed in an Austrian alpine region 1500 m above sea level [16,22,23,24,25,26]. Possible reasons are a complex interplay of abiotic and biotic factors, combined with socio-economic circumstances and anthropogenic factors [26,27,28]. Furthermore, migratory birds may contribute to an expanded occurrence [24,29,30]. In general, TBEV has become an increased international health concern (reviewed in [31]). According to the Centers for Disease Control and Prevention (CDC), several TBE cases in people travelling to Europe, Russia or China were reported during 2000–2009 in the United States of America [32].

Most TBEV infections remain asymptomatic in humans. However, when symptoms occur, patients display a mono- or biphasic course with different degrees of severity depending on the causative TBEV subtype [3,12,33]. Mortality rates vary among the three main subtypes with an increase in fatal cases from TBEV-Eu (1–2%) to TBEV-Sib (6–8%) and to TBEV-FE (up to 40%) (reviewed in [34]). In general, several factors, such as age and immune status of the infected person, infectious dose, TBEV strain and virulence, may influence the severity of the disease (reviewed in [31,35]). Approximately 75% of the symptomatic patients infected with TBEV-Eu display a typical biphasic course. The incubation period after a tick bite, with subsequent transmission of TBEV, ranges from 2–28 days (mainly between 7–14 days) [2,3,12], whereas onset of symptoms after an alimentary transmission is shorter [17]. The disease starts with non-specific symptoms, such as headache, fatigue, nausea or malaise combined with fever. This initial phase lasts for 1–8 days and reflects the viremic period. After an asymptomatic phase of 1–20 days, about one third develop a second phase involving the CNS and manifesting in e.g., meningitis, encephalitis or myelitis. Additionally, the development of long-term or permanent neurological sequelae in TBE patients has been observed [2,3,12]. During the initial (viremic) phase, TBEV RNA is present in the blood and can be detected by reverse transcriptase polymerase chain reaction (RT-PCR). However, patients are often hospitalized after the onset of neurological symptoms and at this time point, the RNA is already cleared from the blood. Thus, RT-PCR has a minor role in the routine diagnostics of TBEV cases. Therefore, TBEV infections are mainly confirmed serologically by the detection of TBEV-specific immunoglobulin (Ig) M and IgG (reviewed in [36]).

Currently, there are no licensed antiviral drugs against TBEV available and treatment of individual patients is supportive only (reviewed in [3]). Apart from preventive measures, such as wearing protective clothes, direct removal of ticks or avoiding consumption of unpasteurized milk (reviewed in [31]), active immunization is the most important protective measure against infection with TBEV.

## 2. Currently Available Vaccines

All licensed vaccines against TBEV are based on inactivated whole viruses, containing various strains of the European or Far-Eastern TBEV subtype (reviewed in [34,37,38]). In general, they can be grouped as European, Russian and Chinese vaccines.

Currently, two European vaccines based on the Austrian isolate Neudoerfl (FSME-IMMUN) and the German isolate K23 (Encepur), both TBEV-Eu strains, are available in many European countries and Canada [34,39,40]. Licensed vaccines in Russia and some neighboring countries are based on the Russian TBEV-FE isolate Sofjin (TBE vaccine Moscow and Tick-E-Vac/Klesch-E-Vac) and Far-Eastern strain 205 (EnceVir) (reviewed in [3,34]). In China, SenTaiBao based on the Chinese TBEV-FE strain Sen-Zhang is approved as a TBEV vaccine (reviewed in [37,38]). For the production of these vaccines, the respective virus isolates are grown in primary chicken embryonic cells (European and Russian vaccines) or primary hamster kidney cells (Chinese vaccine). After virus purification and inactivation, the vaccines are supplemented with an adjuvant, stabilizer and buffer/preservative (reviewed in [3,37]). In general, the vaccines differ in their antigen content and used stabilizer as shown in Table 1. Recently, an aluminum hydroxide-free, inactivated whole virus TBEV vaccine (Evervac, TBEV-FE strain Sofjin) produced in a continuous Vero cell line was tested in a phase I/II clinical trial and showed comparable safety, tolerability and immunogenicity results to the TBE vaccine Moscow. However, this vaccine is not yet licensed [41]. The vaccination schedules of the TBEV vaccines are very time-consuming. Besides the need to administer several doses for primary immunization, booster vaccinations are necessary for maintaining the protective efficacy (Table 1). Apart from the conventional schemes, rapid vaccination schedules for most of these vaccines are available (reviewed in [3,42]). European vaccines can be used interchangeably [43,44].

In general, European and Russian TBEV vaccines are considered safe and well tolerated. However, mild to moderate systemic and local adverse effects, such as fever, headache or redness and swelling at the injection site, can appear both in children and adults [45,46,47,48,49,50,51,52].

All TBEV vaccines are highly immunogenic with high and fast seroconversion rates ranging from 86–100%, depending on the TBEV vaccine, evaluation method and study design. Various studies showed seropositivity levels of 99.1–100% for the European vaccines after complete primary immunization [49], 100% for the Russian vaccines after two immunizations [51,52] and 86–96% for the Chinese vaccine after two vaccinations (reviewed in [37,53]). In general, waning of the vaccine-induced immunity over time has been reported in several studies [49,54,55,56,57].

Despite the broad availability of the European vaccines, vaccine coverage in European countries is relatively low, making TBEV control difficult (reviewed in [3,58]). According to an Austrian field study, efficacy of European vaccines was calculated to be 96–99% in persons with a complete, and 91.3–92.5% in persons with an incomplete vaccination schedule. Due to the high vaccination coverage in Austria, i.e. 85% of the population was vaccinated at least once against TBEV, and the high field effectiveness, it is assumed that approximately 4000 TBE cases were prevented during 2000–2011 [59]. In addition, the extensive vaccination of the Austrian population contributed to a strong decrease in TBE incidences in children, which was 40-times lower compared to the neighboring country Slovenia, although incidence rates were similar before [60]. Furthermore, the beneficial effect of TBEV vaccination was also demonstrated in Russia. Vaccination with TBE vaccine Moscow led to decreased incidence rates and prevention of an estimated annual 1500 novel TBE cases (reviewed in [34]). These facts highlight the importance of vaccination in preventing TBE cases.

However, immunization with TBEV vaccines does not provide complete protection. Consequently, vaccination failures and breakthrough infections can occur in patients with incomplete or even regular TBEV vaccination history. Although all age groups can be affected, breakthrough infections mostly occur in elderly persons older than 50 years of age [61,62,63]. TBE disease severity in patients with vaccination failure (FSME-IMMUN or Encepur) ranges from mild to severe with sometimes long-term neurological sequelae or a fatal outcome [61,63,64,65,66,67]. Between 2000–2015, 1.7% of all reported TBE cases in Slovenia were breakthrough infections [63]. A further study conducted in Stockholm County (Sweden) identified that 5% of all TBE cases between 2006–2015 occurred in vaccinated subjects [62]. However, it is assumed that the case numbers of TBEV breakthrough infections are higher [61,62].

## 3. Immune Response to TBEV Infection and Vaccination

### 3.1. Innate Immunity Against TBEV

The first defense against pathogens is the innate immune response consisting of anatomical and chemical barriers as well as innate immune cells such as natural killer cells, macrophages, neutrophils and dendritic cells. After infection, TBEV-specific pathogen-associated molecular patterns (PAMPs) are recognized by pattern recognition receptors (PRRs) of nucleated cells. Some important PRRs during infection of RNA viruses are Toll-like receptors (TLRs) 3, 7, 8 and 9 or retinoic-acid-inducible gene I (RIG-I)-like receptors. Their activation leads to the production of type I interferon (IFN) (reviewed in [68]), which was shown to have a protective role in TBEV infection (reviewed in [69]). Besides the crucial role of innate immune cells to combat TBEV, there is evidence that some of these cells are infected by TBEV, favor viral spread or contribute to pathogenesis in TBEV infection. Some non-structural proteins of TBEV, such as NS1, NS2A, NS4A, NS4B or NS5, display antagonistic functions, thus, interfering with components of the innate immune response (reviewed in [3,69,70,71,72]; [73]). In addition, TBEV infection modulates expression patterns of many antiviral genes which are involved in the innate immune response such as genes for PRRs, cytokines or chemokines [74]. Besides TBEV itself, tick-derived saliva was shown to modulate the host innate immune response by influencing pathways, such as increasing the activation of the Akt pathway in TBEV-infected dendritic cells [75], and innate immune cells (reviewed in [76]).

However, vaccination aims for the induction of adaptive immune responses and memory functions. Hence, the innate immune response will not be further discussed in this review.

### 3.2. Adaptive Immunity Against TBEV

Virus-specific humoral and cellular immunity, mediated by neutralizing and complement-fixing antibodies as well as T cells with helper, cytotoxic and memory functions, are essential for protection against flavivirus infections. However, in the case of TBEV, it may also pose a risk of increased severity of infection or neuropathology [77].

#### 3.2.1. Antibody Response

Antibodies induced after natural infection and vaccination play an important role as correlates of protection against TBEV since they can prevent (lethal) disease (reviewed in [3]). Antibodies can neutralize TBEV in various manners including prevention of viral attachment or fusion and support of pathogen elimination via the complement system (reviewed in [78]). Therefore, the humoral immune response is critical for controlling virus dissemination, virus clearance and long-term protective immunity (reviewed in [79,80]).

Upon TBEV infection, the amount of TBEV-specific antibodies in human serum and cerebrospinal fluid (CSF) increases and antibodies are usually detectable with the onset of neurological symptoms [2]. The highest IgM levels, which persist for around six weeks, are found in the early stages of TBE, whereas IgG levels peak in the late convalescence period [81]. IgG antibodies can persist for several years and protect from reinfection with TBEV (reviewed in [36]). In general, natural infection induces higher virus-neutralizing (VN) antibody titers than vaccination with an inactivated virus [82]. Low serum levels of VN antibodies combined with a high number of cells (segmented granulocytes (60–70%) and lymphocytes (30–40%)) in the CSF at disease onset are thought to be indicative of a more severe clinical course of TBE [83].

Following primary TBEV vaccination, an increase in antibody titers is observed after the second dose which subsequently decline, followed by a strong booster response upon the third immunization [84]. Studies on long-term persistence of vaccine-induced antibodies showed durability of protective or VN antibodies, respectively, up to 5 years in 99% of children and above 98% in young individuals [54,56,57]. In addition, other studies reported persistence for up to 8 or even 10 years [55,85]. However, with increasing age, immunosenescence, i.e., alteration of the innate and adapted immunity, can be observed in the elderly leading to a lower immunogenicity of vaccines (reviewed in [86]). Noteworthy, avidity and functional activity of TBEV vaccine-induced antibodies are apparently neither influenced by age, age at primary immunization nor last vaccination, but are rather affected by the individual [87].

Several human linear B cell epitopes within the structural and non-structural proteins of TBEV (prM, E, NS1, NS2B, NS3, NS4B and NS5) have been identified. Two of these are located in domain I (DI) and II (DII) of the E protein, respectively, and one in the *C*-terminal RNA-dependent RNA polymerase region of NS5, and showed a positive reaction with almost all positive TBEV sera tested [88]. Intrinsic factors, such as the conformational flexibility (reviewed in [89]) and the maturation state of flaviviruses (reviewed in [90]), as well as extrinsic factors, such as the formaldehyde inactivation of virus particles during the manufacturing process of vaccines (reviewed in [39]; [88,91]), potentially impact sensitivity to antibody-mediated neutralization by changing the accessibility of epitopes. In addition, antibody avidity can be influenced by the addition of adjuvants, e.g., aluminum hydroxide, to the vaccines [92].

Upon TBEV infection and vaccination, epitopes in the E protein are the main target for the induction of VN antibodies (Table 2).

On the viral surface, 180 copies of the E protein structured into 90 dimers can be found in a ”herringbone-like” icosahedral arrangement (reviewed in [109]). The E monomer consists of three distinct domains (DI–DIII) connected by flexible linkers and a membrane anchor domain (DIV) (Figure 1).

DII has an elongated finger-like structure, which is formed by two loops connecting DI, the central part of the protein, in the dimeric E protein. The hydrophobic loop at the tip of DII, covered by DI and DIII in mature virions, is responsible for the fusion of the viral and endosomal membranes, therefore, termed fusion loop (FL) [5]. The sequence of the FL is highly conserved among all flaviviruses. DIII has an immunoglobulin-like structure with exposed loops protruding from the viral surface. The structure of the E protein is influenced by the pH leading to increased exposure of previous inaccessible epitopes, such as the FL [5,111]. All three ectodomains are capable of inducing a VN antibody response (reviewed in [109,112]). Epitopes can comprise not only individual domains of the E protein but also residues from adjacent dimers and dimers in the quaternary herringbone-like arrangement of E proteins at the virion surface [109,113]. In addition, the dynamic behavior of flaviviruses, commonly known as “viral breathing”, can have a great impact on antibody binding (reviewed in [89,109]).

Jarmer et al. [93] provided insights into individual specificity and the variation of the humoral immune response after TBEV infection or vaccination in humans. Using an immunoassay with recombinant E protein and combinations of the single E domains, they observed strong individual variation in antibody titers as well as immunodominance patterns concerning the individual E domains [93]. The human humoral response was dominated by EDI- and EDII-specific antibodies [88,93], whereas in mice immunized with inactivated, aluminum hydroxide-adjuvanted, purified TBE particles mostly EDIII-specific VN antibodies were detected [92]. Furthermore, depletion of antibodies with the dimeric E protein from sera of both naturally infected and vaccinated individuals led to a strong reduction in VN activity, indicating a minor role of complex quaternary epitopes in the VN antibody response. Additionally, depletion of EDI- and EDII-specific antibodies also significantly reduced the VN activity of post-infection or post-vaccination sera, suggesting that antibodies recognizing antigenic sites independent of the dimeric structure of the E protein display considerable VN activity [93]. The characterization of several flavivirus antibodies identified amino acids of the highly conserved FL in EDII as a dominant antigenic site for cross-reactive, but not cross-neutralizing, antibodies [114,115,116]. After solubilization of TBE virions with a mild detergent, EDII-specific antibodies showed a strong increase in binding avidity, indicating that the EDII-FL epitopes of native viruses may have a limited accessibility due to partial occlusion. The fact that these broadly cross-reactive, non-neutralizing antibodies are present in human sera after infection supports the hypothesis that during natural flavivirus infection, cryptic epitopes become accessible and are presented to the host immune system [116].

In addition to the E protein, the TBEV NS1 protein plays an important role as a soluble antigen. In its pentameric or hexameric form, NS1 is secreted into the extracellular space [11], inducing a NS1-specific antibody response. After natural TBEV infection, NS1-specific IgM and IgG antibodies are detectable at high levels in human sera (Table 2) [94,95,96,97]. Several studies showed that NS1 immunization or passive transfer of anti-NS1 antibodies afforded protection against infections with flaviviruses, such as DENV, YFV and WNV in animal models [117,118,119,120,121,122,123]. These findings have been confirmed for the TBEV NS1 protein [97,101,105,106,107,108,124]. Mouse studies showed that immunization with a synthetic peptide corresponding to the structurally conserved α-helix (aa37–55) of NS1 [108] or with various synthetic fragments of NS1 [101] induced partly protective immunity against lethal challenge infection with TBEV. In addition, significantly prolonged survival of TBEV-infected mice was observed after hyperimmunization with a whole recombinant TBEV NS1 protein, although all mice succumbed to infection [97].

Whether or not an anti-NS1 immune response is induced by TBEV vaccination is matter of debate. In some studies, anti-NS1 antibodies were almost exclusively observed after natural infection but not after vaccination [94,95,96]. However, recent studies on the European vaccines question these findings by demonstrating the presence of NS1 in Encepur by mass spectrometry and detection of NS1-specific antibodies by Western blot in sera of vaccinees who repeatedly received Encepur or FSME-IMMUN. Moreover, vaccination with FSME-IMMUN induced a robust anti-NS1 antibody response in mice after several immunizations [97]. Further investigations are needed to confirm these results and to gain a better understanding of the role of vaccine-induced anti-NS1 antibodies. Due to the robust IgM and IgG response, NS1 might be an interesting target for novel vaccine designs and approaches.

In addition to antibodies against the E and NS1 proteins of TBEV, antibodies against the structural prM protein have been described. Analysis of sera obtained from TBE patients and vaccinees showed the presence of prM-specific antibodies, although titers were higher after infection. However, results from depletion studies indicated that prM-specific antibodies play only a minor role in TBEV neutralization [93]. In flavivirus infection, the protective potential of prM-specific antibodies has been shown e.g., for DENV in mice studies [125,126]. In contrast, more recent studies indicated a role of cross-reacting prM antibodies in antibody-dependent enhancement (ADE) in human DENV infection [127,128]. Thus, further research is required to identify which role anti-prM antibodies, induced after infection or vaccination, have in TBEV infection.

##### Impact of Pre-Existing TBEV-Specific Immunity

Once immunization by natural infection or vaccination has taken place, it remains an open question which role pre-existing antibodies may play in further immunizations or TBEV infection. A positive influence of pre-existing TBEV immunity on seroconversion rates following vaccination with Russian TBEV vaccines was observed. However, the effect disappeared 30 days after the first dose was given and after the administration of a second dose [52]. Furthermore, it has been shown that pre-existing TBEV-specific antibodies in immune complexes can have immunomodulatory effects. Immunization of mice with a soluble dimeric E protein only or together with monoclonal antibodies specific for EDI, EDII and EDIII, respectively, revealed differences in the fine specificity of the antibody response. The epitope-specific modulations of the immune response were mechanistically related to the shielding of epitopes in the E protein and the monoclonal antibody-induced dissociation of the E dimer, thereby revealing cryptic epitopes [129].

Due to cross-reactive epitopes, pre-existing vaccine-induced immunity mediates not only protection against homologous, but also heterologous TBEV strains. Among different TBEV subtypes, it was shown that immunization with vaccines based on TBEV-Eu or TBEV-FE elicits cross-subtype reactive antibodies [130,131,132,133,134]. However, studies with various TBEV vaccines showed that cross-protection against heterologous TBEV subtypes is not ensured for any strain [134,135]. Interestingly, one mutation in the EDI/EDII hinge region in the K23 vaccine seed virus (used for manufacturing of Encepur) was identified to be responsible for reduced VN antibody titers against heterologous Neudoerfl virus (vaccine strain used for manufacturing of FSME-IMMUN) of sera received from children vaccinated with Encepur Children. In contrast, sera of children immunized with FSME-IMMUN Junior showed comparable high levels of VN antibodies against both TBEV strains [136].

Antibodies induced by TBEV vaccination afford not only a certain degree of intra-species cross-protection but also cross-react within the TBE complex [133] and with other flaviviruses [116]. Vaccination with FSME-IMMUN or TBE vaccine Moscow protected mice against Omsk hemorrhagic fever virus (OHFV) [130,137]. Furthermore, sera of TBEV vaccinated persons could partially neutralize louping ill virus (LIV) and WNV [138]. Of interest, pre-existing vaccine-induced immunity to YFV prior to TBEV vaccination resulted in an impaired TBEV-specific VN antibody response. Nevertheless, all vaccinated study subjects reached protective TBEV neutralizing antibody titers after completing the recommended TBEV vaccination schedule [84]. On the other hand, pre-existing vaccine-induced immunity to TBEV increased the initial JEV-specific neutralizing antibody response after vaccination with an inactivated JEV vaccine in humans [139]. In general, neutralizing and non-neutralizing antibodies that cross-react with various flaviviruses can be detected post infection and vaccination [116,138,140]. Overall, these results indicate the presence of cross-reactive epitopes among flaviviruses and the effects of pre-existing flavivirus-specific immunity should be taken into consideration when evaluating flavivirus vaccines.

##### Antibody-Dependent Enhancement in TBEV Infection?

Besides the potential protective effects of cross-reactive pre-existing antibodies, they may also have detrimental effects by increasing the severity of disease through ADE of TBEV infection. Non-neutralizing antibodies could promote virus entry into susceptible host cells, leading to increased infectivity. The most common mechanism of ADE involves expression of Fc receptors. Complexes of virus and sub- or non-neutralizing antibodies binding to Fcγ receptors (FcγR) on myeloid cells, such as monocytes and macrophages, increases the attachment and uptake of the virus. As a consequence, this opsonization leads to enhanced infectivity (reviewed in [141]). Haslwanter et al. [142] identified a FcγR-independent mechanism based on interactions of the FL of the TBEV E protein with cellular membranes. Studies with a monoclonal antibody (A5) recognizing an epitope in the E dimer interface demonstrated that binding of A5 triggers the dissociation of E dimers, thus exposing the buried FL. As a consequence, FL-mediated attachment to the plasma membrane increases binding and uptake of the viral particle into the cell, thereby enhancing TBEV infectivity in vitro [142].

Whereas the phenomenon of ADE was extensively described and studied for DENV infection in vitro and in vivo (reviewed in [143,144]), a potential role of ADE in TBEV infections is less well investigated. Early studies showed the ability of polyclonal sera against members of the TBE serocomplex to enhance TBEV replication in vitro [145]. These results are in line with the in vitro observed increased infectivity of TBEV in mouse peritoneal macrophages in the presence of sub-neutralizing TBEV antibodies [146]. Interestingly, it was shown that murine TBEV hyperimmune sera as well as TBEV-specific human polyclonal sera enhanced viral replication in mouse macrophages in vitro, but protected mice from lethal TBEV challenge. Administration of the human or murine immunoglobulins in different dilutions and combined with sublethal challenging doses revealed no signs of ADE in vivo [147]. A more recent study investigating TBEV neutralization in vitro and in mice by treatment with intravenous immunoglobulins containing high amount of TBEV-specific neutralizing antibodies supports these findings since no ADE was observed by the application of cross-reactive or virus-specific antibodies at different neutralization levels [148]. In vivo investigations in mice with sub-neutralizing concentrations of a chimeric antibody against EDIII containing the constant regions of human IgGκ (chFVN145), proposed as post-exposure treatment, also showed no indications of ADE for the three main TBEV subtypes [149]. Additionally, in vivo and in vitro studies addressing the potential effect of TBEV-specific immunity on ZIKV infection suggested that humoral immunity against TBEV is unlikely to contribute to enhancement of ZIKV-induced pathogenesis in humans [150]. In summary, it is not completely clear if ADE plays a role in TBEV infection but it does not seem to be a major problem. ADE has been demonstrated for TBEV in vitro but evidence in vivo is lacking.

#### 3.2.2. CD4^+^ T Cell Response

Virus-specific CD4^+^ T cell responses are essential for the induction of antiviral immunity. CD4^+^ T cells recognize antigenic peptides of 13–25 residues presented by major histocompatibility complex (MHC) class II molecules, which are expressed on antigen-presenting cells (reviewed in [151]). Upon stimulation, they produce pro-inflammatory and antiviral cytokines which results in the recruitment and priming of other lymphoid cells. They may have cytotoxic activity against virus-infected cells and are essential for promoting virus-specific antibody responses (reviewed in [152]).

Against infections with flaviviruses, CD4^+^ T cells play an important role as was demonstrated e.g., for WNV and ZIKV [153,154]. Studies in mice showed that clearance of primary WNV infection from the CNS is dependent on CD4^+^ T cells [154]. Likewise, CD4^+^ T cells contribute to controlling of ZIKV infection and are pivotal for inducing ZIKV-specific antibodies [153]. For TBEV, adoptive transfer of CD4^+^ T cells to TBEV-infected severe combined immunodeficient (SCID) mice, demonstrated a protective role of these T lymphocytes in limiting the development of TBE and mortality [77].

Analysis of the CD4^+^ T cell response to TBEV structural proteins demonstrated that the E and C proteins, but not the prM/M protein, were a major target in infected and vaccinated study subjects (Table 2) [98,99]. Correlation of protein structure and immunodominance patterns revealed that peptides corresponding to helices α2 and α4 of the C protein dominated the CD4^+^ T cell response in both vaccinated and infected individuals, whereas the E protein-specific CD4^+^ T cell response was less focused on selected protein domains. Most E protein-specific CD4^+^ T cells recognized epitopes in β-sheets of EDIII as well as exposed loops protruding from the surface of the viral particle. Additionally, vaccination induced significant EDI- and EDII-specific CD4^+^ T cell responses, whereas CD4^+^ T cell responses against peptides from the stem region of E were predominantly observed in TBE patients [99]. Besides the identified CD4^+^ T cell epitopes in the structural proteins, several predicted T helper cell epitopes were located in the NS1 protein of TBEV [101]. In general, individual variation in CD4^+^ T cell peptide specificity as well as cytokine profile was seen in TBEV naturally infected and vaccinated persons [98,99]. Individuals infected with other flaviviruses, like JEV, revealed a comparable CD4^+^ T cell response, also against the JEV structural proteins and NS1 [155]. Comparison of immunodominance patterns from TBEV-, ZIKV-, DENV- and YFV-specific CD4^+^ T cell epitopes revealed similarities in the distribution of some epitopes. In summary, all four flaviviruses shared epitopes in the two helices α2 and α4 of the C protein and in EDI and EDII, respectively, although some amino acid sequence variation was observed in the respective epitopes [156].

Seven days post hospitalization, an elevated number of activated CD4^+^ T cells with effector functions (measured by higher human leukocyte antigen- (HLA-) DR, PD-1 and perforin expression and decreased Bcl-2 expression) was observed in TBE patients compared to control samples [104]. CD4^+^ T cells responding to DENV, ZIKV, WNV, JEV and YFV are characterized by the production of interferon-gamma (IFN-γ), interleukin-2 (IL-2) and tumor necrosis factor-alpha (TNF-α), indicating a T helper (Th) 1 cell response [153,155,157,158,159,160,161,162]. This Th1 response was also observed after TBEV infection and vaccination but differed in its specificity and functionality [98,99,100,102].

Overall, the frequency of TBEV-specific CD4^+^ T cells was higher after vaccination than after infection [99]. Antagonistic immunomodulatory properties of viral proteins, such as NS5, during viral infection [163] and/or an enhanced T cell response after booster vaccinations may explain these differences. On the other hand, a more polyfunctional phenotype of CD4^+^ memory T cells was found in study subjects who had recovered from acute TBEV infection, compared to vaccinees. In recovered TBE patients, a triple-positive phenotype (IL-2^+^TNF-α^+^IFN-γ^+^) was observed predominantly, while in vaccinated subjects a mono- or bifunctional phenotype predominated (TNF-α^+^ or IL-2^+^TNF-α^+^) [100]. Vaccination induced high levels of IL-2, TNF-α and CD40L expression in CD4^+^ T cells but the IFN-γ response was considerably lower compared to that observed in TBE patients [98,100]. The CD4^+^ T cell response patterns were comparable after all three vaccine doses, with the highest response after the second immunization [100]. The VN antibody response after vaccination correlated with CD4^+^ T cell functions, especially with frequencies of TBEV-specific IL-2^+^ and TNF-α^+^ CD4^+^ T cells [98,99,102]. Whether this is also the case after infection is unclear [98,99].

Studies on cross-reactivity of CD4^+^ T cells specific for mosquito-borne flaviviruses (reviewed in [164]) indicate that e.g., JEV-specific CD4^+^ T cells can recognize other flaviviruses which may result in stronger CD4^+^ T cell responses upon secondary infection with a heterologous virus [165,166]. Data on cross-reactivity of TBEV-specific CD4^+^ T cells with other flaviviruses are lacking.

Overall, knowledge about CD4^+^ T cell response after infection or vaccination is limited. CD4^+^ T cell subpopulations are important for protection against TBEV, however, their correlation with antibody responses and their cross-reactive potential require further evaluation.

#### 3.2.3. CD8^+^ T Cell Response

In addition to virus-specific antibodies and CD4^+^ T cells, CD8^+^ T cells are an important immune correlate of protection against viral infections. CD8^+^ T cells recognize viral peptides of about 8–10 amino acid residues that are presented by MHC class I molecules (reviewed in [151]). So far, seven TBEV CD8^+^ T cell epitopes have been identified, all located in the non-structural proteins (NS2A, NS3, NS4B and NS5; Table 2) [103,104]. Almost all of these epitopes are highly conserved among European, Siberian and Far-Eastern TBEV subtypes [103].

It has been shown that flavivirus-specific CD8^+^ T cells directed to conserved non-structural proteins but also to structural proteins, display cross-reactivity across various flaviviruses. It is unclear if cross-reactive CD8^+^ T cells can afford protection against a heterologous flavivirus or contribute to a more severe disease outcome (reviewed in [164]). Especially for DENV infections this is a matter of debate (reviewed in [167]). However, the potential role of cross-reactive TBEV-specific CD8^+^ T cells is largely unknown.

The peak of CD8^+^ T cell responses in the peripheral blood of acutely infected TBE patients was about seven days after hospitalization during the second, neuroinvasive stage. TBEV-induced effector cells were characterized by increased HLA-DR, PD-1, perforin and granzyme B levels in combination with decreased expression of CD127, Bcl-2 and CD27. Furthermore, most of the CD8^+^ T cells (>50%) showed monofunctional properties in virus control (degranulation, cytokine or chemokine expression) 7, 21 and 90 days after hospitalization. During the course of infection, these cells maintained their monofunctionality but altered their function. The change in the functional composition was accompanied by the differentiation of TBEV-specific CD8^+^ T cells with a dominant Eomes^+^Ki67^+^T-bet^+^ effector phenotype (peak at day seven) into Eomes^−^Ki67^−^T-bet^+^ memory cells as the infection resolved (day 21 and 90) [104].

The MHC haplotype dictates the epitope and T cell repertoire as well as magnitude of the virus-specific CD8^+^ T cell response (reviewed in [151]). This was also demonstrated for TBEV by studies investigating CD8^+^ T cell responses against non-structural TBEV proteins during acute TBEV infection [103,104]. Depending on HLA alleles, the differentiation of effector memory CD8^+^ T cells in two distinct phenotypes was observed. One week after hospitalization, HLA-A2- and HLA-B7-restricted TBEV epitope-specific effector cells showed a common effector memory phenotype (CD45RA^–^CCR7^–^CD27^+^CD57^–^). Two weeks later, mainly effector memory (CD45RA^–^CCR7^–^CD27^+^CD57^+^; approximately 25% of the cells) and terminally differentiated effector memory RA (CD45RA^+^CCR7^–^CD27^+^CD57^+^; approximately 40% of the cells) phenotypes for HLA-A2- and HLA-B7-restricted cells, respectively, were found [103].

In the brain of TBE patients with a fatal disease outcome, TBEV and activated CD8^+^ T cells have been demonstrated [168,169]. Expression of α4β1 integrin (=VLA-4) on CD8^+^ T cells during the neuroinvasive phase of TBEV infection may indicate the migratory capacity of these cells into the CNS [103]. Since TBEV infects neurons of the CNS [169], the recruitment of CD8^+^ T cells may exert, in addition to beneficial effects, detrimental effects and may contribute to the pathogenesis through the killing of infected neurons and the local secretion of cytokines such as TNF-α. This hypothesis is supported by the observation that there were no differences in the extent of virus replication in the brains of mice surviving and in those that succumbed to experimental infection [170]. Since activated TBEV-specific CD8^+^ T cells show upregulation of perforin and granzyme B [104] and CD8^+^ granzyme B^+^ T cells have been found in the brain tissue of fatal TBE cases associated with neuronal damage [168], indeed they may contribute to the pathogenesis, although uncontrolled virus replication may result also in a worse outcome of disease. A correlation between infiltrated CD8^+^ T cells and brain damage was also described for other flaviviruses [171,172,173]. Other evidence for the detrimental effects of virus-specific CD8^+^ T cells stems from work with CD8^–^/^–^ knockout and SCID mice. Upon infection with a neurovirulent TBEV strain, a significantly prolonged mean survival time compared to immunocompetent mice was observed. However, all mice succumbed to infection [77]. Comparable findings have been made for WNV. Infection of CD8^+^-deficient (β2-m^−^/^−^) mice inoculated with a high virus dose resulted in a prolonged mean survival time compared to immunocompetent mice. In contrast, CD8^+^-deficient mice inoculated with a low dose of WNV showed, despite an extended mean survival time, an enhanced mortality rate compared to wild type mice [173]. These findings indicate that during natural flavivirus infections, which are most likely of low virus dose, CD8^+^ T cells may exert beneficial protective effects. Of note, it has been shown that the currently used inactivated TBEV vaccines induce virus-specific CD8^+^ T cell responses inefficiently, if at all [100,174,175].

In summary, CD8^+^ T cells are thought to contribute to protective immunity and recovery from TBEV infection. However, further research should rule out potential downsides of CD8^+^ T cell-mediated immunity.

#### 3.2.4. Vaccine Failures

Understanding the cause of breakthrough infections is very important for improving existing vaccines and developing new ones. Although a common definition of vaccination breakthrough infection is missing, TBE patients developing disease despite being vaccinated indicate the inability of the vaccine to induce a protective immune response in any case [176]. Host-related risk factors, such as advanced age (beyond 60 years), as well as vaccine-related risk factors, such as incomplete primary immunization or delayed first booster doses, may favor vaccination breakthrough infections (reviewed in [177]). Regarding TBEV vaccination, two types of vaccination breakthrough infections have been suggested: patients with inadequate immune response and non-responders [176].

The humoral immune response upon natural infection of individuals responding to TBEV vaccination but lacking protective efficacy, differs from those of unvaccinated TBE patients. In vaccinated but unprotected study subjects, IgG and VN antibodies increased rapidly, but the IgM response was delayed [61,63,64,65,66,67,178,179]. However, this differential serological profile does not explain the lack of protection.

Comparison of TBEV-specific immune responses post booster vaccination in high and non-responders revealed significant differences in their antibody response. In the non-responder group, TBEV-specific geometric mean titers (GMT) of VN antibodies were below the detection limit pre-booster (6.5 GMT). A slight increase two months after booster vaccination (18.0 GMT) with a decline six months later (10.7 GMT) was observed. In contrast, high responders showed ten-fold higher VN antibody titers pre-booster (67.5 GMT) which strongly increased eight weeks after booster vaccination (139.6 GMT) and decreased slightly afterwards (101.1 GMT) [180]. Investigation of TBEV-specific IgG avidity in both groups revealed no differences, suggesting alterations in quantity rather than quality of antibodies [63,67,179,180]. Further studies showed a correlation of humoral and cellular immunity in both groups: low antibody titers were associated with poor antigen-specific T cell proliferation, whereas high antibody titers were associated with strong T cell proliferation. Upon TBEV antigen-restimulation, peripheral blood mononuclear cells from TBEV vaccine non-responders were characterized by low levels of IL-2 and IFN-γ production prior and post booster vaccination, contrary to high responders with higher cytokine expression levels. Furthermore, a decrease in regulatory T cells was observed in the high responder group after the booster vaccination, while non-responders displayed an increase in IL-10-producing regulatory T cells. Consequently, this might disable T cell proliferation or impair B cell-induced IgG production. This increase observed in non-responders is thought to be due to the expansion of natural as well as inducible regulatory T cells. On the other hand, the decrease in high responders may be explained by the expansion of TBEV-specific effector T cells, changing the overall distribution of CD4^+^ subpopulations. In addition, no correlation between HLA class II haplotype and vaccination failure in TBEV non-responders was found. Interestingly, TBEV vaccine non-responders were able to develop an efficient immune response upon vaccination with seasonal influenza vaccines (Inflexal V 2008/2009 and Inflexal V 2009/2010), as observed in high responders. Hence, the non-responsiveness to TBEV vaccines is supposed to be antigen-dependent at humoral levels [180].

To overcome impaired vaccine responsiveness, especially in the elderly, further research on the understanding of TBEV vaccination breakthrough infections is warranted. Optimized or novel vaccine strategies are needed to ensure sufficient protection for all risk groups.

## 4. Novel Approaches and TBEV Target Antigens for the Development of Improved TBEV Vaccines

The currently available TBEV vaccines suffer from some disadvantages as discussed above, including: (a) the time-consuming vaccination regimen and need of regular booster vaccinations for maintaining the protective efficacy, (b) reduced immunogenicity in the elderly as well as (c) vaccination breakthrough infections and non-responders (vaccine failure). Therefore, the development of improved vaccines against TBEV is desirable. An ideal TBEV vaccine should fulfill several requirements:
Highly immunogenic in all age and risk groups, rapid and high seroconversion rates.Induction of long-lasting immunity without the need for booster vaccinations.No vaccine failures.Protection against all TBEV subtypes.Cost-effective and safe.

To fulfill these requirements, various aspects must be considered. Central to the development of improved TBEV vaccines is obtaining a better understanding of the correlates of protection and consequently identifying the targets for humoral and cellular immune responses. In addition, the vaccine platform chosen for delivery of the antigen(s) of interest can influence the nature of the induced immune response. Thereby, combining different arms of the immune system is desirable to induce a strong/protective and long-lasting immunity. Although various vaccine platforms and antigens were already utilized for designing novel TBEV vaccine approaches, data on safety, tolerability and efficacy in men are lacking.

### 4.1. Novel TBEV Vaccine Strategies Aiming at the Induction of Humoral Immunity

The humoral arm of the adaptive immunity plays a major role in protection against TBEV. Of note, low VN antibody titers were associated with a more severe TBE outcome [83]. Thus, a vaccine should induce high affinity VN antibodies which prevent infection, dissemination and support viral clearance.

For TBEV it is known that VN antibodies are mainly directed to epitopes located in the E protein (Table 2). However, immunization with various soluble forms of dimeric whole or membrane anchor-free TBEV E protein failed to induce VN antibodies and provide appropriate immunogenicity in mice [181]. Likewise, DNA vaccines driving intracellular expression of either whole or truncated TBEV E proteins or secretion of TBEV E dimers, failed to induce VN antibodies and afford robust protection against TBEV [182]. Thus, the usage of TBEV E protein only was not very successful. In addition, flavivirus subunit vaccines based on a single protein domain of the E protein, namely EDIII, were tested. EDIII is a biologically relevant and flavivirus-specific target, inducing antibodies that are less cross-reactive than those directed against epitopes in EDI/EDII, thus, reducing a possible risk of ADE [115,144,183]. Immunization of mice with an attenuated recombinant influenza A virus (rIAV) encoding the WNV EDIII protein domain in frame with the IAV neuraminidase protein, induced VN antibodies and CD4^+^ T cells as well as protection against lethal WNV challenge [184]. Contrary, immunization of mice with EDIII of ZIKV expressed from a plasmid, replication-deficient chimpanzee adenovirus or given as a recombinant protein confirmed that EDIII of ZIKV induced VN antibody responses inefficiently [185]. However, results from mouse studies need to be evaluated carefully since TBEV immunodominant regions were shown to differ in humans (EDI and EDII) [93] and mice (EDIII) [92]. Since EDIII seems to play a minor role in eliciting VN antibodies upon TBEV infection in humans [93], it is unclear whether EDIII would be a promising target for novel TBEV vaccines.

In addition to vaccine candidates based on the TBEV E protein only, those based on the E and prM proteins have been tested extensively. Co-expression of E and prM results in the self-assembly of protein complexes without a genome, known as virus-like particles (VLPs). Thus, VLPs are unable to replicate and their use is considered to be safe (reviewed in [186]). In vitro expression of prM and E in a VLP expression system based on mammalian cell lines [187,188] or the yeast *Pichia pastoris* [189], resulted in the production of such recombinant TBEV VLPs. These VLPs were smaller than native whole virions but showed similar surface protein organization (including antigenic structures) and functional reactivity (rearrangement and fusion at low pH, hemagglutination activity) when compared to native virus particles [190]. Due to their authentic morphology, which mimic native virus particles, VLPs are highly immunogenic and able to provoke humoral as well as cellular immune responses (reviewed in [186,191]). Immunization of mice with purified VLPs produced in mammalian cells led to a VN antibody response and complete protection against TBEV challenge. However, immunogenicity and protective efficacy were similar to formalin-inactivated whole TBE viruses [181]. Thus, improvements to achieve a stronger immunogenicity, superior to that after vaccination with inactivated whole viruses, is desirable. One opportunity might be the incorporation of C proteins in the VLPs as demonstrated for ZIKV VLPs. Direct comparison of recombinant ZIKV VLPs without (prM-E) and with C (C-prM-E), both produced in a stable cell line, showed that in mice immunized with C-prM-E VLPs neutralizing antibody titers were enhanced, C-specific antibodies were detectable and viremia upon challenge infection was prevented [192]. In addition to recombinant VLPs produced in vitro, they can also be produced in situ after vaccination with nucleic acid-based vaccines or live attenuated viruses encoding the prM and E proteins (Figure 2).

Immunization with a DNA-based vaccine enabling assembly of recombinant TBEV VLPs in vivo, elicited the production of VN antibodies in mice [182,193] and rhesus macaques [194]. Additionally, immunized mice or mice treated with serum of vaccinated non-human primates were protected against TBEV challenge [182,193,194]. Moreover, in situ generation of TBEV VLPs by vaccination of mice with a late-defective recombinant Vaccinia virus (rVACV) encoding for the prM and E proteins, induced VN antibodies and a robust protection upon challenge infection with TBEV [195]. Altogether, these data suggest that co-expressed prM and E induced superior immune responses with higher protective efficacy than the E protein alone. Therefore, strategies including prM and E should be pursued in further research.

Besides VN antibodies against the E protein, anti-NS1 antibodies were detectable in sera of TBE patients [94,95,96,97]. Mouse studies showed that these antibodies afforded partial protection from TBEV infection [97,101,108]. Therefore, the NS1 protein may be considered a promising additional vaccine target in future TBEV vaccines. Immunization with a replication-defective recombinant adenovirus (rAd) encoding the NS1 protein of TBEV elicited NS1-specific antibodies in mice, prevented viremia and afforded partial protection [106,107]. The NS1-induced protective immunity is mediated mainly by antibodies and B cells as was demonstrated by adoptive transfer experiments [196]. Similar results were obtained with rVACV encoding for TBEV NS1, which afforded partial protection against TBEV challenge infection [197]. Although NS1-based TBEV vaccine preparations may not be suitable as stand-alone vaccines, its addition to vaccines might increase their efficacy. Studies with an attenuated recombinant vesicular stomatitis virus (rVSV) encoding for a polyprotein consisting of ZIKV prM, E and NS1 support this idea. Immunization of mice with this live attenuated vaccine candidate elicited high levels of VN antibodies, cellular immunity and complete protection against ZIKV challenge infection [198]. Although, evidence for ADE of TBEV in vivo is lacking, the potential risk of enhanced infectivity should be taken into consideration when developing novel vaccines. Thus, the use of the NS1 protein as vaccine antigen might be of interest since it can contribute to protective immunity with a presumably low risk of ADE, which could be an issue for the E protein.

As shown for some of the licensed TBEV vaccines [130,131,132,133,134,135], ideally novel TBEV vaccines should induce cross-protective immunity to other TBEV subtypes. This has been achieved with vaccine candidates containing the prM-E or NS1 protein. Gene Gun immunization of mice with infectious TBEV RNA carrying a large deletion in the 3’ non-coding region (NCR), which allowed attenuated replication and expression of all viral proteins, afforded protection against a lethal dose of a heterologous TBEV strain [199]. Additionally, protection against heterologous TBEV strains in mice was achieved by transfer of serum obtained from non-human primates immunized with a TBEV prM-E DNA vaccine [194]. Moreover, the cross-protective potential against heterologous TBEV subtypes as well as against OHFV, another member of the *Flaviviridae* family, was demonstrated in mice with a rAd encoding NS1 of TBEV [196]. Likewise, cross-protective antibodies against Langat virus (LGTV), a flavivirus closely related to TBEV (reviewed in [200]), were induced in mice after immunization with plasmid DNA encoding the TBEV prM and E proteins [193]. Interestingly, a modified lipid nanoparticle-mRNA vaccine encoding prM and E of Powassan virus (POWV), a member of the TBE serocomplex, induced antibodies that cross-reacted with TBEV and LGTV and protected mice from LGTV challenge infection, despite these three flaviviruses displaying only 77% amino acid identity in the E protein [201].

To summarize, data on TBEV or other flavivirus vaccine candidates based on various vaccine platforms and proteins demonstrated their potential in eliciting VN antibodies and affording protection against challenge infections. Due to the importance of VN antibodies in protective immunity to TBEV, their induction by future vaccines is a must. To our knowledge, none of the vaccine candidates described above have reached clinical trials yet. Thus, data on immunogenicity and protective efficacy rely on pre-clinical studies and must be confirmed clinically.

The decline of TBEV-specific antibody titers necessitates regular booster vaccinations (Table 1) to maintain protection afforded by the licensed vaccines [49,54,55,56,57], which sometimes fail to provide protection, especially in the elderly [61,62,63,64,65,66,67]. The induction of robust cellular immunity by novel vaccine candidates may help to prevent these problems and potentiate protective efficacy.

### 4.2. Novel TBEV Vaccine Approaches Aiming at the Induction of Cellular Immunity

Various novel TBEV vaccine candidates have been tested for their capacity to enhance immunity by induction of virus-specific T cell responses. The importance of virus-specific T cells in protective immunity to TBEV infection has been demonstrated by adoptive transfer experiments to naïve recipient mice [77]. In humans, the VN antibody levels induced after vaccination correlated with CD4^+^ T cell functionality [98,99,102]. For the induction of long-lasting protective immunity, quantity but also quality of CD4^+^ T cells are of importance since these T lymphocytes regulate B cell responses and antibody production. On the one hand, follicular helper T (Tfh) cells are necessary to produce high affinity antibodies and on the other hand, CD4^+^ T cells contribute to protective immunity by e.g., differentiation into cytokine-producing polyfunctional cells (Th1 response) and establishment of an immunological memory (reviewed in [202,203]). TBEV CD4^+^ T cell epitopes (Table 2) were identified within the E and C proteins [98,99]. Predictions suggested the presence of epitopes in the TBEV NS1 protein [101]. Comparable immunodominant epitopes in E and C were identified for ZIKV, DENV, YFV and JEV [155,156]. T cells directed to these epitopes might display cross-reactivity. Thus, TBEV vaccines may induce a certain degree of cross-reactive immunity to other flaviviruses.

With live attenuated rIAV and modified Vaccinia virus Ankara (rMVA) as WNV vaccine candidates, WNV EDIII and E protein specific CD4^+^ T cell responses, respectively, could be induced [184,204]. Also the use of flavivirus VLPs led to the induction of virus-specific CD4^+^ T cell responses, as was shown for ZIKV VLPs [205,206]. Comparison of mouse CD4^+^ T cell responses induced with rVSV encoding the ZIKV prM-E alone or together with the NS1 protein showed that co-expression of NS1 increased CD4^+^IFN-γ^+^ T cell responses and decreased CD4^+^TNF-α^+^ Th1 cells [198]. This underlines that NS1 has apparently immune modulatory properties and can alter the nature of the T cell response.

Besides the used proteins, the antigen delivery system and the route of administration can have a major impact on vaccination outcome. For example, comparison of intramuscular vaccination of mice with DNA (TBEV prM-E) induced a Th1 response (more IgG2a than IgG1 antibodies, dominant IFN-γ response), whereas Gene Gun immunization provoked a Th2 response (more IgG1 than IgG2a antibodies, dominant IL-4 and IL-5 response) [182]. Furthermore, the magnitude of the VN antibody response was influenced by the application method of another DNA-based vaccine (TBEV prM-E). Mice immunized intramuscularly by a needle-free jet injector developed a greater VN antibody response than after subcutaneous vaccination with needle-syringe injection, albeit lower amounts of DNA were used for needle-free immunization. The wide dispersion and therefore enhanced uptake of DNA after needle-free administration most likely account for this difference [207]. Using nucleic-acid vaccines, the biochemical delivery method and use of adjuvants influence vaccine-induced immunity, too (reviewed in [208,209]). With the use of adjuvants, immune responses can be biased, like in the case of MF59, to enhanced Tfh and germinal B cell responses [210].

Since virus-specific CD8^+^ T cells also contribute to protective immunity against virus infections, proteins with CD8^+^ T cell epitopes are considered as vaccine targets. For TBEV, CD8^+^ T cell epitopes in the non-structural proteins NS2A, NS3, NS4B and NS5 have been identified [103,104]. Immunization of mice with a rVACV expressing TBEV C-prM-E-NS1-NS2A-NS2B-NS3 (vC-NS3) demonstrated that including non-structural proteins, which contain CD8^+^ T cell epitopes, increased the protective efficacy of this vaccine preparation (Figure 3). Immunization with vC-NS3 induced superior TBEV-specific antibody levels, limited viral load most efficiently and provided the highest level of protection compared to immunization with rVACV encoding C-prM-E-NS1 or 5′NCR-C-prM-E-NS1-NS2A of TBEV only [211].

Upon vaccination with live viruses, antigens are synthesized in the cytoplasm of infected cells which facilitates endogenous antigen processing and subsequent MHC class I-restricted presentation of viral peptides. Consequently, the induction of virus-specific CD8^+^ T cell responses are facilitated. For example, WNV antigen-specific (memory) CD8^+^ T cell responses were promoted in mice with a rMVA-WNV vaccine candidate [204]. In contrast, inactivated vaccines, like the licensed TBEV vaccines, induce CD8^+^ T cells inefficiently [100,174,175,212]. One of the most effective and safest vaccines used in humans is the live attenuated YFV 17D vaccine. This vaccine induces VN antibodies and T cell responses which provide long-term protective immunity against Yellow Fever after a single dose (reviewed in [213]). Such a durable vaccine-induced immunity is desirable since repeated vaccinations are not required.

For TBEV, a live virus vaccine based on an attenuated LGTV strain was evaluated in Russia by the immunization of approximately 650,000 volunteers. A single vaccination led to 100% seroconversion in all vaccinees and antibodies persisted for several years. However, this LGTV-based vaccine proved not to be safe and caused a relatively high incidence of encephalitis. Furthermore, the vaccine failed to afford complete protection against TBEV, probably because of limited antigenic cross-reactivity between LGTV and local TBEV strains (reviewed in [200]). Although these first attempts to use LGTV as a live virus vaccine against TBEV were overshadowed by severe neurological adverse events, further investigations were undertaken to develop safe, attenuated TBEV vaccine strains that mimic TBEV infection upon vaccination. Deletions or introduction of several mutations in the 3′NCR, C, E or NS5 proteins were identified to contribute to an attenuated TBEV phenotype [199,214,215,216,217,218,219,220]. These approaches opened new opportunities for producing attenuated live virus vaccines that allow the induction of antibody and T cell responses to both structural and non-structural TBEV proteins. Subsequent studies in mice showed that such attenuated TBE viruses provided protection against challenge infection with wild type TBEV [214,216,217,218,219,220]. However, to our knowledge none of these candidates have been further evaluated in detail or tested in clinical trials, likely due to safety concerns.

Further studies investigated live attenuated chimeric flaviviruses for their capacity to induce protective immunity with a safe profile. Currently, two live chimeric flavivirus vaccines, the tetravalent vaccine Dengvaxia against DENV (reviewed in [221]) and IMOJEV against JEV (reviewed in [222]), based on the attenuated YFV 17D backbone are licensed for use in humans. For TBEV, flavivirus chimera based on various backbones derived from YFV 17D, attenuated JEV, DENV-2, DENV-4, LGTV and the WNV RepliVax platform, have been designed by replacing the prM and E genes of the respective virus vector with those of TBEV or LGTV [223,224,225,226,227,228,229,230,231,232]. Although, TBEV-specific antibodies and protective immunity were induced, for several of those chimeric viruses neurovirulence constituted a safety issue [223,224,233,234]. On the other hand, promising results regarding safety and protective efficacy were obtained in mice and non-human primates with the RepliVax WNV/TBEV [223], a microRNA-targeted TBEV/LGTV [231] and a LGTV/DENV-4 chimera [227,233,235,236]. Interestingly, a single dose of the microRNA-targeted TBEV/LGTV vaccine candidate induced VN antibody titers in non-human primates comparable to those after three doses of Encepur [231]. The chimeric LGTV/DENV-4 vaccine candidate, however, induced VN-cross-reactive TBEV-specific antibodies inefficiently in human study subjects. Booster vaccination failed to enhance the cross-reactive TBEV-specific antibody levels [237]. Collectively, these studies highlight the difficulties finding the optimal balance between immunogenicity and attenuation/safety of chimeric viruses. Moreover, vaccines based on chimeric flaviviruses are dependent on cross-reactive T cells between the chosen flavivirus backbone and TBEV, since TBEV-specific T cell epitopes are mainly located in the C and non-structural proteins which are encoded by the backbone of the vaccine. Of note, it was recently shown that CD4^+^ and CD8^+^ T cells induced by the YFV 17D vaccine exhibited limited cross-reactivity with other mosquito-borne flaviviruses as well as lower antigen-sensitivity against heterologous antigens [238]. These findings might suggest that with e.g., TBEV/YFV 17D chimera as a novel vaccine candidate, inadequate TBEV-specific T cell responses could be evoked. Therefore, the protective efficacy of cross-reactive B and T cell responses to TBEV have to be addressed in further studies.

To include TBEV non-structural genes in the vaccine formulation and circumvent safety issues of live attenuated virus vector vaccines, a replication-competent, non-infectious TBEV RNA was generated. By introducing an in-frame deletion of approximately two-thirds of the C protein, production of infectious virions was prevented, whereas point mutations in the prM signaling sequence promoted efficient release of subviral particles [174,239]. Upon immunization, TBEV-specific VN antibodies, Th1 and CD8^+^ T cell responses were induced, quantitatively and qualitatively comparable to those induced by vaccination with a live attenuated virus. Pre-existing immunity to TBEV seemed to not have negative effects because humoral and CD8^+^ T cell responses increased after booster immunization [174]. Of interest, the virus harboring the deletion in the C protein only and lacking the point mutations in the prM signaling sequence, failed to induce VN antibody titers. Nevertheless, all mice were protected against lethal TBEV challenge infection [239]. This highlights that cellular mediated immunity is an important correlate of protection and supports the notion that inducing T cell responses to non-structural proteins should be considered a favorable property of future TBEV vaccine formulations.

For the use of viral vectors, interference by pre-existing vector immunity is considered a potential disadvantage (reviewed in [240]). The use of heterologous prime-boost vaccination schemes may overcome this problem and evoke durable memory T cell responses (reviewed in [241]). By priming e.g., with a DNA vaccine, especially the cellular immune response to the antigen of interest can be increased upon booster vaccination with a vaccine of another nature (reviewed [242]), like viral vectors based on poxviruses [243] or adenoviruses [244,245]. These favorable vaccination outcomes have been demonstrated in mouse studies using TBEV NS1-encoding rVACV for priming and TBEV NS1-encoding plasmid DNA for boosting. This heterologous prime-boost strategy provided protective immunity against lethal TBEV challenge infection, whereas several repeated vaccinations were needed with each of the individual vaccine preparations to achieve this [246].

## 5. Conclusions and Future Perspectives

Over the last few years, TBEV has become a major health concern due to the increased spread of TBEV endemic foci and ticks, as well as rise in TBE incidences (reviewed in [21]). The most efficient protective measure against TBE is vaccination. Despite the good tolerability, safety and field efficacy of currently available inactivated TBEV vaccines, they exhibit some shortcomings, such as the need for time-consuming vaccination schedules or incomplete protection, especially in the elderly. Novel vaccination strategies may overcome some of the limitations of the licensed TBEV vaccines. For the development of novel vaccines, thorough understanding of the immune correlates of protection against TBEV is crucial and should be the subject of further investigations. Efficient induction of virus-specific memory B and T cell responses is pivotal for providing durable protective immunity and the prevention of vaccination breakthrough infections in TBEV vaccinees. In addition to long-lasting TBEV E and NS1 protein-specific protective antibodies, robust TBEV-specific cellular immune responses will contribute to favorable disease or vaccination outcomes. So far, a limited number of CD4^+^ T cell epitopes in the structural [98,99,100], and CD8^+^ T cell epitopes in the non-structural proteins [103,104] of TBEV have been identified. However, the contribution of T cell responses to these individual epitopes affording a protective immunity has been under-investigated and the full repertoire of T cell specificities is largely unknown. Likewise, the immune response elicited by the licensed TBEV vaccines is still not fully understood.

Issues that need to be addressed include the induction of NS1-specific antibodies upon vaccination with the licensed vaccines [94,95,96,97] and the cross-reactivity of TBEV-specific T cells. So far, in vivo proof of ADE for TBEV is lacking, however, the effects of pre-existing immunity against other flaviviruses in this regard should be kept in mind when developing novel vaccine approaches.

A number of novel TBEV vaccination strategies are in various stages of development and have been tested, mainly in pre-clinical models (Table 3). Combinations of TBEV structural and non-structural antigens have been tested for their capacity to induce protective VN antibodies and virus-specific CD4^+^ and CD8^+^ T cell responses. To our knowledge, a live attenuated LGTV/DENV-4 chimera has been the only novel vaccine candidate tested in a clinical trial. Although it proved to be highly attenuated and safe, cross-reactivity of the antibody response with TBEV was poor [237]. In general, most vaccination strategies mainly include the prM and E proteins of TBEV to elicit protective antibody responses. The inclusion of TBEV proteins that induce potent CD4^+^ and CD8^+^ T cell responses may improve vaccine efficacy and durability. Moreover, further testing of the NS1 protein as vaccine antigen seems warranted since it afforded partial protection in mouse models.

A better understanding of the correlates of protection against TBEV obtained in recent years combined with lessons learned from novel vaccine strategies against TBEV or other flaviviruses may pave the road for the development of improved TBEV vaccines that can be safely used in all age and risk groups and that afford solid protection from TBE.

## Figures and Tables

**Figure 1 vaccines-08-00451-f001:**
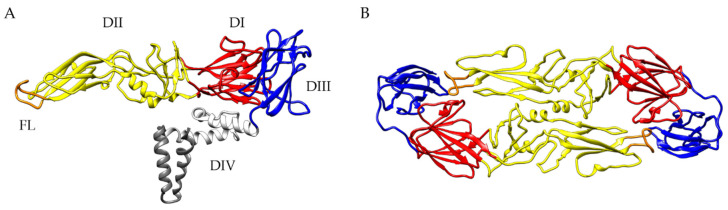
The structure of the TBEV E protein (PDB-ID 5O6A [5]). (**A**) Side view of a single TBEV E protein monomer. Depicted are the four domains (DI: red, DII: yellow, DIII: blue, DIV (stem/anchor): gray) and the fusion loop (FL: orange). (**B**) Top view of a soluble TBEV E protein dimer. Color code same as in (**A**). Ribbon diagrams were prepared with UCSF Chimera [110].

**Figure 2 vaccines-08-00451-f002:**
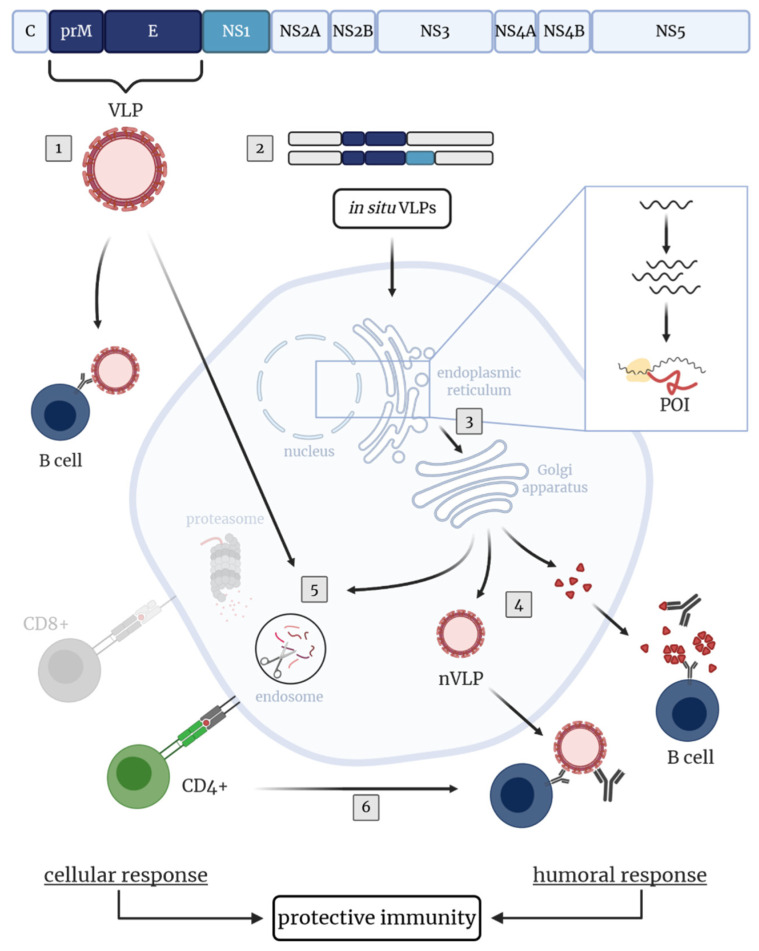
TBEV-specific immunity upon immunization with virus-like particles (VLPs). The TBEV polyprotein with proteins inducing protective antibody responses (prM, E, NS1) is shown. Upon immunization with purified VLPs (1) or vector-based vaccines co-expressing prM and E (in situ production of VLPs, (2)), mainly humoral immune responses are induced. Inclusion of the NS1 protein in these vector-based vaccine approaches may improve the protective immunity. (3) Vector-based immunization leads to the synthesis of proteins of interest (POIs), which are processed on different pathways leading to humoral and/or cellular immune responses. (4) Humoral immune response: POIs are either secreted into the extracellular space (NS1) or assemble into novel VLPs (nVLPs; prM-E) which are subsequently released from the cell. The secreted proteins and de novo produced (nVLPs) or directly administered VLPs, respectively, induce the production of TBEV-specific B cells and antibodies. (5) Cellular immune response: The POIs or endocytosed VLPs, respectively, are degraded into peptides mainly by host proteases in endosomes leading to antigen presentation via MHC class II molecules to CD4^+^ T cells. (6) CD4^+^ T cells promote the activation and proliferation of TBEV-specific B cells driving efficient antibody responses with development of memory responses. Created with BioRender.com.

**Figure 3 vaccines-08-00451-f003:**
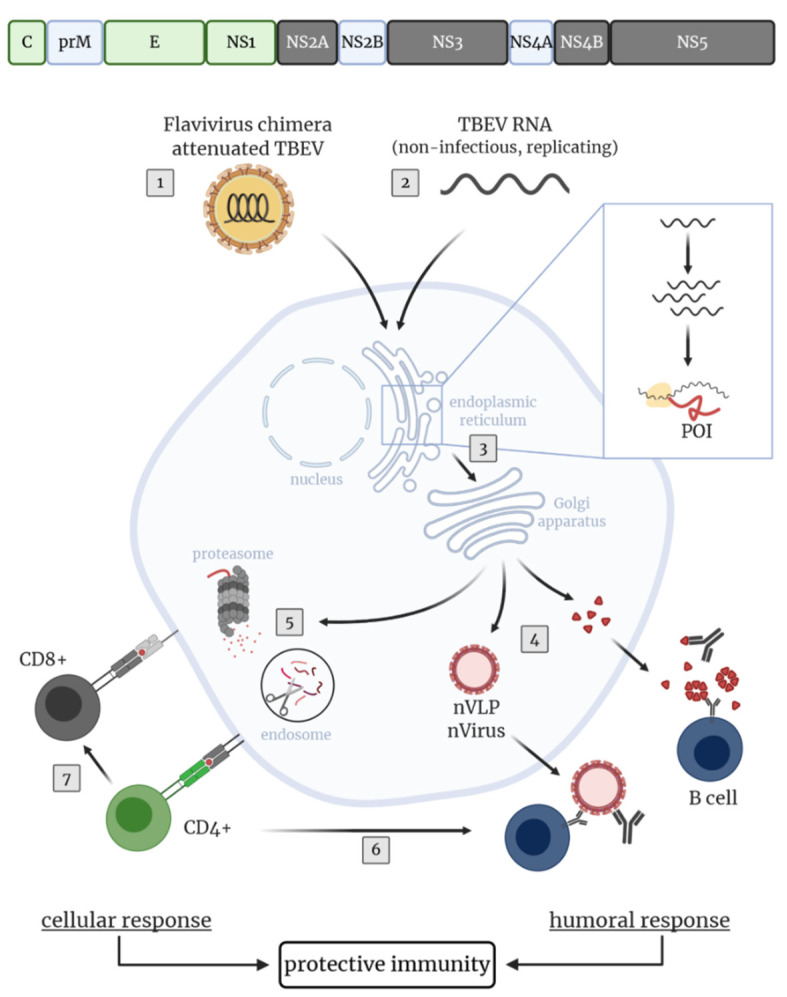
Inclusion of flavivirus non-structural proteins to enhance vaccine-induced immunity. The TBEV polyprotein with proteins inducing CD4^+^ (C, E, NS1) and CD8^+^ (NS2A, NS3, NS4B, NS5) T cell responses is shown. Upon immunization with live attenuated flaviviruses (1) or non-infectious, replicating TBEV RNA (2), structural as well as non-structural proteins are included, promoting a humoral but also an efficient cellular immunity. (3) Immunization leads to the synthesis of proteins of interest (POIs), which are processed on different pathways leading to humoral and/or cellular immune responses. (4) Humoral immune response: POIs are either secreted into the extracellular space (NS1) or assemble into novel VLPs (nVLPs) or virions (nVirus) which are subsequently released from the cell. Thus, TBEV-specific B cells and antibodies are induced. (5) Cellular immune response: The POIs are degraded into peptides either by host proteases in endosomes or by proteasomes in the cytoplasm. By this process antigens are presented via MHC class II or I molecules to CD4^+^ or CD8^+^ T cells. CD4^+^ T cells promote the activation and proliferation of TBEV-specific B cells (6) and CD8^+^ T cells (7) with the development of memory responses. Created with BioRender.com.

**Table 1 vaccines-08-00451-t001:** Approved TBEV vaccines and immunization schedules.

Vaccine [3]	TBEV Strain (Subtype) [3]	Antigen Content [3]	Adjuvant [3]	Stabilizer [3]	Pediatric Vaccine Available [3]	Immunization Schedule [42]
FSME-IMMUN ^a^	Neudoerfl (TBEV-Eu)	2.4 µg	Al(OH)_3_	HSA	Yes	1st + 2nd dose: 1–3m, 3rd dose: 5–12m, 1st booster dose: after 3y, subsequent booster doses every 5y (<60 years) * or every 3y (≥60 years)
Encepur ^b^	K23 (TBEV-Eu)	1.5 µg	Al(OH)_3_	Sucrose	Yes	1st + 2nd dose: 2w–3m, 3rd dose: 9–12m, 1st booster dose: after 3y, subsequent booster doses every 5y (<60 years) * or every 3y (≥60 years)
TBE vaccine Moscow ^c^	Sofjin (TBEV-FE)	1.0 ± 0.5 µg/mL	Al(OH)_3_	Sucrose, HSA, gelatose	No, used for ≥3 years	1st + 2nd dose: 1–7m, 1st booster dose: after 1y, subsequent booster doses every 3y
Tick-E-Vac ^c^	Sofjin (TBEV-FE)	1.0 ± 0.5 µg/mL	Al(OH)_3_	Sucrose, HSA	Yes	1st + 2nd dose: 1–7m, 1st booster dose: after 1y, subsequent booster doses every 3y [3]
EnceVir ^d^	205 (TBEV-FE)	2.0–2.5 µg	Al(OH)_3_	Sucrose, HSA	Yes	1st + 2nd dose: 1–7m, 1st booster dose: after 1y, subsequent booster doses every 3y
SenTaiBao ^e,^ [37]	Sen-Zhang (TBEV-FE) [37,38]	n.k.	Al(OH)_3_ [37]	HSA [37]	No, used for ≥8 years [37]	1st + 2nd dose: 1–2w, annual booster doses

Manufacturer: ^a^: Pfizer/USA, ^b^: GlaxoSmithKline plc/United Kingdom, ^c^: Chumakov FSC R&D IBP RAS/Russia, ^d^: Microgen-Branch FSUC “SIC “Microgen” of MOH of Russia “SIC “Virion”/Russia, ^e^: Changchun Institute of Biological Products Co., Ltd./China. *: Germany: 50 years instead of 60 years as age limit. Abbreviations: Al(OH)_3_: aluminum hydroxide, Eu: European, FE: Far-Eastern, HSA: human serum albumin, m: months, n.k.: not known, TBEV: tick-borne encephalitis virus, w: weeks, y: year(s).

**Table 2 vaccines-08-00451-t002:** TBEV proteins eliciting TBEV-specific immune responses in humans upon TBEV infection or TBEV vaccination.

	**Infection**	**Vaccination**
**Protective antibodies**	E [88,93] NS1 [94,95,96,97]	E [88,93] NS1 [97] ^#†^
**CD4+ T cells**	C [98,99,100] E [98,99,100] NS1 [101]	C [98,99,100] E [98,99,100,102]
**CD8+ T cells** *	NS2A [103] NS3 [103,104] NS4B [103] NS5 [103]	-

^†^: protective potential was shown in mouse models only [97,101,105,106,107,108]. ^#^: presence of NS1-specific antibodies in vaccinees still under discussion [94,95,96,97]. *: CD8+ T cell epitopes were identified for TBE patients with HLA-A2 and HLA-B7 haplotypes [103,104].

**Table 3 vaccines-08-00451-t003:** In vivo evaluated approaches for TBEV vaccine candidates and their outcome.

Approach	Strategy	Included TBEV Target Antigens	TBEV-Specific Adaptive Immunity			Protection (Challenge Virus)	Comment	Ref.
			Antibodies	CD4+ T Cells	CD8+ T cells			
Vaccination with proteins								
Mammalian cell line-based expression system	Production of recombinant E protein (soluble dimeric E without membrane anchor) from plasmid (COS-1 cells), comparison to virus-derived E dimers (with/without membrane anchor) and E rosettes (multimeric aggregates) Evaluated in mice	E (Dimers or rosettes)	+(VN-Ab)	n.d.	n.d.	++/+(TBEV)	E dimers: Very low VN-Ab Low or no protection against heterologous TBEV strain E rosettes: Higher VN-Ab titer compared to E dimers Complete protection against heterologous TBEV strain	[181]
**Virus-like particles (VLPs)**								
Mammalian cell line-based expression system	Production of VLPs from recombinant plasmid (COS-1 cells), purified VLPs used for immunization Evaluated in mice	prM-E	+(VN-Ab)	n.d.	n.d.	++(TBEV)	Protection against heterologous TBEV strain Higher VN-Ab titer compared to E rosettes (see above) VN-Ab and protection levels comparable to formalin-inactivated whole virus (used as control)	[181]
**DNA vaccines**								
DNA encoding for VLPs	In vivo production of VLPs from plasmid DNA encoding prM-E Evaluated in mice and NHP	prM-E	+(VN-Ab in mice + NHP)	+	n.d.	++(TBEV: mice)	Mice: CD4+ T cell response: Th1/Th2 response depending on administration method (i.m. or Gene Gun) Complete protection against homo- and heterologous TBEV strains Protection comparable to inactivated vaccine (FSME-IMMUN) NHP: VN-Ab cross-reactive to heterologous TBEV strains Passive transfer of NHP sera protected mice from TBEV challenge infection VN-Ab titer and passive protection comparable to inactivated vaccine (FSME-IMMUN)	[182,193,194]
DNA encoding for E protein	Immunization with plasmid DNA encoding antigens (secreted terminally truncated soluble E dimer, non-secreted full-length E, inefficiently secreted truncated E) Evaluated in mice	E	+(VN-Ab)	+	n.d.	+/−(TBEV)	Very low VN-Ab (depending on antigen) CD4+ T cell response: Th1/Th2 response (depending on administration method (i.m. or Gene Gun)), weaker compared to VLPs or inactivated vaccine (FSME-IMMUN)	[182]
**RNA vaccines**								
‘Naked’ infectious RNA	Application of infectious in vitro synthesized RNA of an attenuated TBEV mutant (carrying a 470 nt deletion in the 3′NCR for attenuation), immunization with purified infectious RNA leading to replication of highly attenuated mutant virus in vivo Evaluated in mice	Whole TBEV	+	n.d.	n.d.	++(TBEV)	Immunization afforded protection against heterologous TBEV strain Seroconversion rate comparable to non-attenuated parental virus	[199]
‘Naked’ non-infectious RNA	Application of in vitro synthesized non-infectious, replication-competent TBEV RNA (carrying an in-frame deletion of aa28–89 in the C protein with or without three point mutations (Gly112Pro; Met113Gln and Leu115Gln)) Evaluated in mice	Whole TBEV	+(VN-Ab)	+	+	++(TBEV)	Inactivated vaccine (FSME-ImmunInject) failed to elicit a CD8+ T cell response Without point mutations: No induction of VN-Ab but complete protection against heterologous TBEV strain With point mutations: Induction of long-lasting VN-Ab, Th1 and CD8+ T cell responses VN-Ab and CD8+ T cells increased after booster vaccination Complete protection against heterologous TBEV strain IgG titer comparable to inactivated vaccine (FSME-ImmunInject) CD8+ response and protection level comparable to attenuated live TBE viruses (CΔ28–43 and 3′NCRΔ10378–10847)	[174,239]
**Recombinant adenoviruses (rAds)**								
Human rAd	Insertion of TBEV NS1 under control of CMV major immediate-early promoter into replication-deficient Rad51ΔE1 Evaluated in mice	NS1	+	n.d.	n.d.	+(TBEV)		[106,107]
Human rAd	Insertion of TBEV NS1 into Rad51 Evaluated in mice	NS1	+	+	+	+(TBEV, OHFV)	Partial protection against heterologous TBEV strains and OHFV Adoptive transfer of serum, B or T cells provided partial protection (for T cells after pre-treatment with cyclophosphane only) CD4+ T cell response: Th1 Administration of Rad51-NS1 together with inactivated TBEV vaccine potentiated the immune response	[196]
**Recombinant Vaccinia viruses (rVACV)**								
VACV	Insertion of TBEV NS1 into thymidine kinase gene under control of early–late poxvirus P65 promoter into VACV Evaluated in mice	NS1	+	n.d.	n.d.	+(TBEV)	TBEV challenge dose dictated protective efficacy	[197]
VACV	Insertion of prM-E into a non-replicating late defective VACV (Uracil DNA glycosylase deficient) Evaluated in mice	prM-E	+(VN-Ab)	n.d.	n.d.	++/+(TBEV)	Protection against TBEV challenge infection depended on dose and route of immunization (s.c. immunization more efficient than i.m. and i.p.)	[195]
VACV	Insertion of structural and non-structural TBEV genes into thymidine kinase gene under control of VACV 7.5k promoter into VACV (C-prM-E-NS1 (vC-NS1); 5‘NCR-C-prM-E-NS1-NS2A (v5‘C-NS2A); C-prM-E-NS1-NS2A-NS2B-NS3 (vC-NS3)) Evaluated in mice	C-prM-E-NS1/5‘NCR-C-prM-E-NS1-NS2A/C-prM-E-NS1-NS2A-NS2B-NS3	+(VN-Ab)	n.d.	n.d.	++/+(TBEV)	vC-NS3: Highest Ab titers and complete protection from challenge infection Level of protection comparable to inactivated TBEV vaccine	[211]
VACV/DNA	Prime-boost vaccination with VACV and bacterial plasmid expressing TBEV NS1 (recombinant VACV: NS1 into thymidine kinase gene under control of synthetic early-late poxvirus promoter; bacterial plasmid: NS1 under control of CMV immediate-early promoter) Evaluated in mice	NS1	+	n.d.	n.d.	+(TBEV)	Partial protection against heterologous TBEV strain Higher protective efficacy upon VACV prime/plasmid boost compared to plasmid prime/VACV boost	[246]
**Live attenuated viruses**								
LGTV	Administration of attenuated LGTV strain Evaluated in humans	Whole LGTV	+	n.d.	n.d.	+(field study)	Tested in 649,470 volunteers Single immunization induced long-lasting seroconversion in 100% of individuals Reduced incidence of TBEV in endemic regions High incidence of severe neurological reverse advents No absolute protection against TBEV in endemic regions	[200]
Attenuated TBEV	Attenuation of TBEV by introducing deletions in the variable 3′NCR region Evaluated in mice	Whole TBEV	+	n.d.	n.d.	++(TBEV)	Genetically stable, attenuated phenotype Immunization afforded protection against heterologous TBEV strain i.c. inoculation of suckling mice showed remaining neurovirulence (slight reduction of neurovirulence; manifested in longer mean survival time)	[218]
Attenuated TBEV	Introduction of single or multiple mutations in EDIII (aa308–311), combination of mutations in the 3′NCR with mutations at EDIII aa310 Evaluated in mice	Whole TBEV	+	n.d.	n.d.	++(TBEV)	All substitutions yielded viable viruses Appearance of compensatory second-site mutations leading to phenotypic reversion Immunization with E(Thr310Lys)3′(∆10847) and E(Thr310Lys)3′(∆10919) afforded protection against heterologous TBEV challenge Substitution Thr310Leu significantly reduced neuroinvasiveness Less efficient attenuation approach compared to [218]	[217]
Attenuated TBEV	Multiple passaging of TBEV in BHK-21 cells and selection of binding mutants with high heparin sulfate affinity Evaluated in mice	Whole TBEV	+	n.d.	n.d.	++(TBEV)	E protein mutants Glu201Lys, Glu122Gly, Ser158Arg/Gly159Arg showed reduced neuroinvasiveness and afforded protection against heterologous TBEV challenge	[219]
Attenuated TBEV	Introduction of deletions into the TBEV C protein (4–21 aa deletions starting at aa28 of C) Evaluated in mice	Whole TBEV	+	n.d.	n.d.	++(TBEV)	Highly attenuated phenotype Introduction of deletions led to VLP production Mutant C(∆28–43) showed reduced neuroinvasiveness and afforded protection against heterologous TBEV challenge	[216]
Attenuated TBEV	Introduction of deletions into the TBEV C protein (19, 21, 27 or 30 aa deletions starting at aa28 of C) Evaluated in mice	Whole TBEV	+	n.d.	n.d.	++(TBEV)	Appearance of compensatory second-site mutations in the C protein upon passaging Second-site mutation Gln70Leu or duplication Ile78-Leu85 restored viability of the parental virus C(∆28–48) C(∆28–48/Gln70Leu) and C(∆28–48/Du78–85): Highly attenuated phenotype Protection against heterologous TBEV challenge Seroconversion rate superior compared to C(∆28–43)	[220]
Attenuated TBEV	Large-scale random codon re-encoding, random introduction of 273 synonymous mutations into NS5 Evaluated in mice	Whole TBEV	+(VN-Ab)	n.d.	n.d.	++(TBEV)	Compared to wild type virus: attenuated phenotype, reduced neurovirulence, reduced neuroinvasiveness, comparable VN-Ab levels	[214]
**Flavivirus chimera**								
JEV-based	Replacement of prM-E from JEV live vaccine strain SA14-14-2 by corresponding genes from TBEV (ChinTBEV) Evaluated in mice	prM-E	+(VN-Ab)	n.d.	n.d.	++/+(TBEV)	Attenuated phenotype but remaining neurovirulence Protection against TBEV challenge infection depended on dose and route of immunization (i.p. or s.c.)	[224]
YFV 17D-, DENV-2- or LGTV-based	Replacement of prM-E from YFV 17D, DENV-2 or LGTV by corresponding genes from TBEV Evaluated in mice	prM-E	+(VN-Ab)	+	n.d.	++/+(TBEV)	Reduced neuroinvasiveness but high neurovirulence (under-attenuation) VN-Ab titer higher in YFV/TBEV than DENV-2/TBEV vaccinated mice Protection levels differed among immunization with chimera (YFV/TBEV: 100% survival, DENV-2/TBEV: 85.7% survival) CD4+ T cell response: Th1 (YFV/TBEV)	[223]
RepliVax (RV) platform	Replacement of prM-E from different flaviviruses (WNV, LGTV, TBEV or YFV 17D) by corresponding genes from TBEV, attenuation due to deletion in C Evaluated in mice and NHP	prM-E	+(VN-Ab in mice + NHP)	+	n.d.	++/+(TBEV: mice, LGTV: NHP)	Highly attenuated in mice RV-WNV/TBEV: Suitable for heterologous prime-boost vaccination with inactivated vaccine (FSME-IMMUN) Only moderate effect of WNV pre-existing immunity Mice: Protection levels differed among immunization with chimera (RV-WNV/TBEV and RV-TBEV/TBEV: 100% survival, RV-YFV/TBEV: 50% survival) VN-Ab titer highest for RV-WNV/TBEV and lowest for RV-YFV/TBEV CD4+ T cell response: Th1 (RV-WNV/TBEV) NHP: Protection depended on dose and application method (i.d., i.m. or s.c.) RV-WNV/TBEV higher immunogenicity and more durable immunity compared to inactivated vaccine (FSME-IMMUN)	[223]
LGTV-based	Replacement of prM-E of DENV-4 with corresponding genes from LGTV (LGTV/DENV-4) Evaluated in humans (phase I trial)	prM-E	+(VN-Ab against LGTV + TBEV)	n.d.	n.d.	n.d.	Good safety profile High seroconversion rates against LGTV (80%) but low against TBEV (35%) Low cross-reactive Ab response against TBEV, no increase observed post booster vaccination	[237]
DENV-4-based	Replacement of prM-E of DENV-4 with corresponding genes from TBEV (TBEV/DENV-4) Evaluated in mice and NHP	prM-E	+(VN-Ab in NHP; n.d. for mice)	n.d.	n.d.	+(LGTV)	Mice: High neurovirulence (suckling mice), reduced neuroinvasiveness (SCID mice) NHP: High VN-Ab titer against TBEV and LGTV (ratio 5:1) VN-Ab titers higher than induced by TBEV/DENV-4∆30 (see below) Booster vaccination enhanced VN-Ab VN-Ab titer against TBEV lower compared to immunization with inactivated TBEV vaccine (Encepur) Protection against viremia upon LGTV challenge infection No attenuated phenotype	[233]
DENV-4-based	Replacement of prM-E of DENV-4 with corresponding genes from TBEV and introduction of a 30 nt deletion in the DENV-4 3′NCR (TBEV/DENV-4∆30) Evaluated in mice and NHP	prM-E	+(VN-Ab in NHP; n.d. for mice)	n.d.	n.d.	+(LGTV)	Mice: High neurovirulence (suckling mice), reduced neuroinvasiveness (SCID mice) NHP: VN-Ab against TBEV and LGTV (ratio 3.5:1) VN-Ab titers lower compared to LGTV/DENV-4 Booster vaccination enhanced VN-Ab VN-Ab titer against TBEV lower than induced by inactivated TBEV vaccine (Encepur) Significantly attenuated phenotype but still neurovirulent (histopathological lesions) Protection against viremia upon LGTV challenge infection	[233,234]
DENV-4-based	Introduction of single or multiple mutations into TBEV/DENV-4∆30 (deletion of 30 nt in 3′NCR of DENV-4; mutations in TBEV E (aa315) and DENV-4 NS5 (aa654, aa655)) Evaluated in mice	prM-E	n.d.	n.d.	n.d.	n.d.	TBEV/DENV-4∆30 was further attenuated in its neurovirulence, neuroinvasiveness and neuro-pathology Mutations are genetically stable in mice Mutant with multiple mutations (vΔ30/E315/NS5654,655) showed highest attenuation level of neuroinvasiveness and neurovirulence	[232]
miRNA targeted flavivirus chimera	Replacement of DENV-4 prM-E by corresponding genes from TBEV and introduction of single or multiple miRNA targeting sequences for cellular CNS-specific miRNAs into the 3′NCR of TBEV/DENV-4 Evaluated in mice and NHP	prM-E	+(VN-Ab in mice + NHP)	n.d.	n.d.	++/+(Parental TBEV/DENV-4: mice)	Occurrence of virus-escape mutants (reversion of neurovirulence), improvement of genetic stability by inserting multiple miRNA targeting sequences Mice: Significant reduction of neurovirulence and neuroinvasiveness Survival rate after challenge infection depended on applicated miRNA targeted TBEV/DENV-4 NHP: VN-Ab response after one dose higher than after three doses of inactivated TBEV vaccine (Encepur)	[228], [229]
miRNA targeted flavivirus chimera	Replacement of LGTV prM-E by corresponding genes from TBEV and introduction of multiple miRNA targeting sequences for cellular CNS-specific miRNAs into the C, NS1 and 3′NCR of TBEV/LGTV Evaluated in mice and NHP	prM-E	+(VN-Ab in mice + NHP)	n.d.	n.d.	++(Parental TBEV/LGTV: mice, NHP; TBEV: mice)	Genetically stable Restricted neuropathogenicity in mice and NHP Mice: Protection against parental chimera and heterologous TBEV strain NHP: Highly immunogenic after one dose VN-Ab response after one dose comparable to that after three doses of inactivated TBEV vaccine (Encepur)	[231]

Legend: ++: complete protection, +: partial protection/presence of antibodies/CD4^+^ T cells/CD8^+^ T cells, -: no protection/absence of antibodies/CD4^+^ T cells/CD8^+^ T cells, n.d.: not determined. Abbreviations: Δ: deletion, aa: amino acid(s), Ab: antibodies, Ad: adenovirus, Arg: arginine, BHK-21 cells: baby hamster kidney cells, C: capsid protein, CMV: cytomegalovirus, CNS: central nervous system, COS-1 cells: green monkey kidney cells, DENV-2/-4: dengue virus serotype 2/4, Du: duplication, E: envelope protein, Gln: glutamine, Glu: glutamic acid, Gly: glycine, i.c.: intracranial, i.d.: intradermal, IgG: immunoglobulin G, Ile: isoleucine, i.m.: intramuscular, i.p.: intraperitoneal, JEV: Japanese encephalitis virus, Leu: leucine, LGTV: Langat virus, Lys: lysine, Met: methionine, miRNA: microRNA, NCR: non-coding region, NHP: non-human primates, NS: non-structural protein, nt: nucleotides, OHFV: Omsk hemorrhagic fever virus, prM: precursor-membrane protein, Pro: proline, rAd/Rad: recombinant adenovirus, Ref.: reference, RV: RepliVax, s.c.: subcutaneous., SCID: severe combined immunodeficient, Ser: serine, TBEV: tick-borne encephalitis virus, Th: T helper cell, Thr: threonine, VACV: Vaccinia virus, VLP: virus-like particle, VN: virus-neutralizing, WNV: West Nile virus, YFV: Yellow fever virus.

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
