# Peer review of "Tick-Borne Encephalitis Virus: A Quest for Better Vaccines against a Virus on the Rise"

_vaccines, 2020, doi:10.3390/vaccines8030451_

Round 1

Reviewer 1 Report

Manuscript Number: Vaccines-886883

Title: Tick-Borne Encephalitis Virus: A Quest for Better Vaccines Against a Virus on the Rise

Article Type: Review

Comments to the authors:

Summary:

The review discusses the adaptive immune response induced upon TBEV infection and vaccination and discusses novel approaches for an improved vaccine. I thoroughly enjoyed reading the review and am certain that it will be highly cited. I highly recommend the publication of this article in Vaccines after consideration of a few minor comments below:

L32: or West Nile Virus – I would write it as “and West Nile Virus” and perhaps also include Zika virus. The authors also forgot to define ZIKV, which they later use in the text.

L44-45: Rewrite to something like “The prM protein forms heterodimers with the E protein, thereby protecting the E protein fusion loop from premature fusion during flavivirus maturation.” I think the terminology is “Fusion loop” and I am not aware of fusion peptide.

L192-193: Do we know why natural infection induces higher virus-neutralizing Ab titers?

L219: Can the authors include a reference (I think Plevka lab) on the high resolution structure of TBEV while discussing about the structure.

L221: DII has an elongated finger-like…… sounds better

L224-225: “The sequence of the fusion loop is highly conserved……” sounds better

Table 2: Can this be reformatted. It is a bit confusing

L369: “adaptive” instead of adoptive

L471-472: just use “natural flavivirus infections, which most likely …..” and remove “of man”

Figures 1 and 2: Is it possible to label the components in 3 and 5

A general comment. It might improve the readability of the paper, if the authors prepared a figure of the TBEV virion structure and labelled the domains in the E protein as they keep discussing about domains I, II and III. As a flavivirologist, I understand the terminology and structure. My comment is meant for a broader audience. I leave the decision to the authors to include or not to include a structure figure.

Author Response

We thank the reviewer for taking the time to read our manuscript and the helpful comments. We have modified the manuscript accordingly as follows:

L32: or West Nile Virus – I would write it as “and West Nile Virus” and perhaps also include Zika virus. The author also forgot to define ZIKV, which they later use in the text.

We included Zika virus in the list of human-pathogenic flaviviruses at the beginning and introduced the abbreviation for Zika virus (ZIKV) in the text. Moreover, we changed “or” to “and”. The sentence is now “The genus Flavivirus comprises several human-pathogenic arthropod-borne viruses such as yellow fever virus (YFV), dengue virus (DENV), Japanese encephalitis virus (JEV), Zika virus (ZIKV) and West Nile virus (WNV).” (page 1, line 35-37).

L44-45: Rewrite to something like “The prM protein forms heterodimers with the E protein, thereby protecting the E protein fusion loop from premature fusion during flavivirus maturation.” I think the terminology is “Fusion loop” and I am not aware of fusion peptide.

According to the suggestion of the reviewer, we changed “fusion peptide” into “fusion loop” and re-wrote this sentence more clearly. The new sentence is now “The prM protein forms heterodimers with the E protein, thereby protecting the E protein fusion loop from premature fusion during flavivirus release [7].” (page 2, lines 48-49).

L192-193: Do we know why natural infection induces higher virus-neutralizing Ab titers?

The commercial TBEV vaccines are all based on inactivated viruses and therefore, the viruses contained in the vaccines are not able to replicate within the host. In contrast, natural infection leads to replication of the infectious virus and e.g. an induction of T cell responses will take place. For example, CD4+ T cell responses are induced which promote efficient B cell responses. Novel TBEV vaccines should aim to induce also T cell responses as discussed in chapter 4. On page 16, lines 676-679 we describe the beneficial role of induction of a CD4+ T cell response as follows “For the induction of long-lasting protective immunity, quantity but also quality of CD4+ T cells are of importance since these T lymphocytes regulate B cell responses and antibody production. On the one hand, follicular helper T (Tfh) cells are necessary to produce high affinity antibodies […]”. To highlight this, we added to the sentence the information that the vaccination contains inactivated virus. The sentence is now “In general, natural infection induces higher virus-neutralizing (VN) antibody titers than vaccination with an inactivated virus [82].” (page 5, line 200-201).

L219: Can the authors include a reference (I think Plevka lab) on the high resolution structure of TBEV while discussing about the structure.

The mentioned publication (Füzik et al., 2018) is already cited in the paragraph about the structure of the TBEV E protein (page 7, line 247, reference [5]). We now cited this paper again earlier in the paragraph (page 7, line 244), so that interested readers find this publication more easily and can have a look on the high-resolution structural figures of the TBEV E protein. Moreover, we considered your general comment about including a structural figure (please see the last comment) but we generated just figures of the TBEV E monomer and dimer since this is the main focus of antibody binding sites.

L221: DII has an elongated finger-like…sounds better.

We changed “prolonged finger-like” into “elongated finger-like” as proposed and the sentence is now “DII has an elongated finger-like structure, which is formed by two loops connecting DI […]” (page 7, lines 241).

L224-225: “The sequence of the fusion loop is highly conserved…” sounds better.

We changed “almost completely” in “highly conserved” as proposed. The re-written sentence is now “The sequence of the FL is highly conserved among all flaviviruses. […]” (page 7, lines 244).

Table 2: Can this be reformatted. It is a bit confusing

The reviewer is right that the table 2 looks confusing and cluttered. Originally, there was more space/paragraphs in the table. We changed the format back to the original format and would kindly ask the editor to not change the table format due to reader-friendliness reason (page 6, line 227 et seqq.).

L369: “adaptive” instead of adoptive

The authors of the cited publication (reference [77], see below) performed adoptive transfer studies in mice. That is the reason why we kept “adoptive” to maintain the meaning (page 9, lines 384-387).

  1. Růžek, D.; Salát, J.; Palus, M.; Gritsun, T.S.; Gould, E.A.; Dyková, I.; Skallová, A.; Jelínek, J.; Kopecký, J.; Grubhoffer, L. CD8+ T-cells mediate immunopathology in tick-borne encephalitis. Virology 2009, 384, 1–6, doi:10.1016/j.virol.2008.11.023.

L471-472: just use “natural flavivirus infections, which most likely…” and remove “of man”.

We changed the wording of the sentence as proposed and it is now “These findings indicate that during natural flavivirus infections, which are most likely of low virus dose, CD8+ T cells may exert beneficial protective effects.” (page 11, lines 486-488).

Figures 1 and 2: Is it possible to label the components in 3 and 5?

It is a good suggestion to label the components 3 and 5 in these two figures. We have labeled the cellular components endoplasmic reticulum, Golgi apparatus, endosome and proteasome (page 14  and page 17).

A general comment. It might improve the readability of the paper, if the authors prepared a figure of the TBEV virion structure and labelled domains in the E protein as they keep discussing about domains I, II and III. As a flavivirologist, I understand the terminology and structure. My comment is meant for a broader audience. I leave the decision to the authors to include or not to include a structure figure.

Indeed a structural figure showing the TBEV E protein with the different domains improves the comprehensibility of the readers. We generated a novel figure displaying an TBEV E monomer and a soluble E dimer. The figure is included in the text where the structure of the E protein is described (page 6).

Reviewer 2 Report

This is a very comprehensive review of Tick-borne encephalitis immunology and vaccine strategies. It is well-organized and clearly written, and does an excellent job of summarizing recent findings on immune correlates of protection.  I have no major criticisms of this work, and feel it should be published.

One addition that may be of interest to readers is inclusion of any known information on the role of tick salivary proteins in the innate immune response to the pathogen. 

Author Response

We thank the reviewer for taking the time to assess our manuscript and the objective evaluation. The impact of tick-derived salivary proteins on the innate immune response is of importance. We have added a paragraph to address this issue in section 3.1  (page 5, lines 179-181):

“Besides TBEV itself, tick-derived saliva was shown to modulate the host innate immune response by influencing pathways, such as increasing the activation of the Akt pathway in TBEV-infected dendritic cells [75], and innate immune cells (reviewed in [76]).”

The cited references are:

75.       Lieskovská, J.; Páleníková, J.; Langhansová, H.; Chmelař, J.; Kopecký, J. Saliva of Ixodes ricinus enhances TBE virus replication in dendritic cells by modulation of pro-survival Akt pathway. Virology 2018, 514, 98–105, doi:10.1016/j.virol.2017.11.008.

76.       Kotál, J.; Langhansová, H.; Lieskovská, J.; Andersen, J.F.; Francischetti, I.M.B.; Chavakis, T.; Kopecký, J.; Pedra, J.H.F.; Kotsyfakis, M.; Chmelař, J. Modulation of host immunity by tick saliva. J. Proteomics 2015, 128, 58–68, doi:10.1016/j.jprot.2015.07.005.